# The root of anomalously specular reflections from solid surfaces on Saturn's moon Titan

Jason D. Hofgartner [1✉], Alexander G. Hayes[2], Donald B. Campbell[2], Jonathan I. Lunine [2], Gregory J. Black [3], Shannon M. MacKenzie [4], Samuel P. D. Birch [2], Charles Elachi[5], Randolph D. Kirk[6], Alice Le Gall[7,8], Ralph D. Lorenz [4] & Stephen D. Wall[1]

Saturn's moon Titan has a methane cycle with clouds, rain, rivers, lakes, and seas; it is the only world known to presently have a volatile cycle akin to Earth's tropospheric water cycle. Anomalously specular radar reflections (ASRR) from Titan's tropical region were observed with the Arecibo Observatory (AO) and Green Bank Telescope (GBT) and interpreted as evidence for liquid surfaces. The Cassini spacecraft discovered lakes/seas on Titan, however, it did not observe lakes/seas at the AO/GBT anomalously specular locations. A satisfactory explanation for the ASRR has been elusive for more than a decade. Here we show that the ASRR originate from one terrain unit, likely paleolakes/paleoseas. Titan observations provide ground-truth in the search for oceans on exoearths and an important lesson is that identifying liquid surfaces by specular reflections requires a stringent definition of specular; we propose a definition for this purpose.

[1] Jet Propulsion Laboratory, California Institute of Technology, Pasadena, CA, USA. [2] Department of Astronomy, Cornell University, Ithaca, NY, USA. [3] Department of Astronomy, University of Virginia, Charlottesville, VA, USA. [4] Applied Physics Laboratory, Johns Hopkins University, Laurel, MD, USA. [5] Division of Geological and Planetary Science, California Institute of Technology, Pasadena, CA, USA. [6] Astrogeology Science Center, United States Geological Survey, Flagstaff, AZ, USA. [7] LATMOS/IPSL, UVSQ Université Paris-Saclay, UPMC Univ. Paris 06, CNRS, Guyancourt, France. [8] Institut Universitaire de France, Paris, France. ✉email: Jason.D.Hofgartner@jpl.nasa.gov

A nomalously specular radar reflections from the southern tropical region of Saturn's moon Titan (equator to ≈27°S, Saturn and Titan have a solar obliquity of ≈27°) were observed with the Arecibo Observatory and Green Bank Telescope from 2000-2008 and interpreted as evidence for liquid surfaces[1,2]. The Cassini Saturn orbiter (e.g., ref. [3]) imaged Titan's surface at infrared and microwave wavelengths from 2004-2017 and discovered >500 lakes/seas[4]. It did not, however, observe liquid surfaces in the regions that are anomalously specular to AO/GBT[2]. Invoking transient liquids is one possible resolution of this apparent discrepancy as methane precipitation does occur on Titan[5]; a more satisfactory explanation has been elusive.

Here we show that the AO/GBT ASRR are correlated to one terrain unit and that this terrain unit has both smoother surfaces and a greater dielectric constant (different composition) than its surroundings, both of which contribute to the anomalous reflections. Transient liquids are not necessary to explain the ASRR. We argue that the terrain unit is likely paleolakes/paleoseas (sites of former lakes/seas), a geomorphologic unit that is of interest for both geology and astrobiology and is all the more interesting given its unique properties relative to other geomorphologic units on Titan. Our conclusion that Titan's ASRR originate from solid surfaces, suggests that to identify liquids on exoearths by specular reflections, a stringent definition of specular should be used. We recommend a definition based on the coherence of the reflected electromagnetic waves.

## Results

### Arecibo Observatory and Green Bank Telescope anomalously specular radar reflections

Titan's surface is not resolved in AO/GBT radar observations, however, the Doppler shift from Titan's rotation results in an echo spectrum that is equivalent to the scan of a slit across Titan's disk[1,2]. Figure 1a–e show example AO/GBT echo spectra of Titan (from the 83 observations, acquired over eight oppositions, in Black et al.[2]). The central, 1-Hz wide, Doppler bin corresponds to a ≈14-km wide slit that includes the entire ≈5150 km (Titan's diameter) height of the disk. Higher (or lower) frequency bins correspond to slits shifted laterally away from the center of Titan's disk in the direction of the limb rotating toward (or away from) Earth. The spectra are in units of normalized radar cross section (NRCS), which is the ratio of the radar power backscattered to the receiver to the power that would have been received if the power incident on the surface had been scattered isotropically; effectively, NRCS is the brightness in radar observations (e.g., ref. [6]). The NRCS of a surface generally decreases with increasing incidence angle (e.g., refs. [2,7,8]). As a result, AO/GBT spectra generally decrease away from the central Doppler bin (slit over the center of the disk), because the central Doppler bin includes the subradar point, where the incidence angle is zero degrees.

The transmitted signal in the AO/GBT observations was circularly polarized and the echo power was measured in both the opposite and same circular senses as that transmitted[1,2]. A specular reflection would originate from the subradar point and increase the NRCS of the central Doppler bin of the opposite circular polarization spectrum. Thus, for the purpose of investigating the AO/GBT ASRR, we use the maximum NRCS of each AO/GBT opposite circular polarization spectrum of Titan (NRCS of the specular component of the spectrum). The AO/GBT ASRR are the observations with anomalously high maximum-NRCS. We use the maximum-NRCS rather than the surface root-mean-square slope parameter of a radar scattering model that was used in previous analyses[1,2] because Titan's complex surface is variable over many length scales. The model parameter depends on neighboring Doppler bins, so it is

contaminated by longitudinal surroundings. The maximum-NRCS depends on only the central Doppler bin and thus has a greater weighting from the terrain at the subradar location[2].

Figure 1f shows histograms of the maximum-NRCS of the AO/GBT observations from Black et al.[2]. The noise in the AO/GBT spectra varied substantially between observations (e.g., Fig. 1a–e). To distinguish between observations with a high maximum-NRCS that may be due to greater noise and those that are confidently anomalous, Fig. 1f includes both the histogram for all of the AO/GBT observations (black) and the histogram for only the observations where the maximum-NRCS was ≥4 times the standard deviation of the noise (red). Two observations have a conspicuously greater maximum-NRCS than all other observations and thus are distinctly anomalous (they are also the two most specular observations using the surface root-mean-square slope parameter instead of maximum-NRCS). Another group of observations with maximum-NRCS of ≈1.5-2 are intermediate; their maximum-NRCS is greater than most other observations but also substantially less than the two extreme ASRR. We consider these observations also to be anomalously specular and argue below that they originate from the same terrain unit as the two distinctly anomalous observations.

### Arecibo Observatory, Green Bank Telescope, and Cassini observations

The subradar locations of the AO/GBT observations are shown on a map of Titan's surface in Fig. 2a, b. The AO is limited to observing zenith angles <≈20° and thus the radar system can only observe Titan during ≈1/3 of the oppositions over Saturn's ≈29.5-year orbital period (i.e., from 2000 to 2008 but not 2009 to 2027). The AO-Titan geometry further limits the subradar locations to a latitude range of ≈20° within Titan's southern tropical region (i.e., from 7 to 27°S). The GBT shares these limitations because it does not have a radar transmitter and its radar observations of Titan were restricted to receiving echoes from the AO transmitter.

Figure 2 is similar to Fig. 17 in Black et al.[2], with four important distinctions. Firstly, the size of the AO/GBT dots indicates the maximum-NRCS rather than the surface root-mean-square slope parameter of a radar scattering model (discussed above). Secondly, the spin models (rate and pole) used to determine locations on Titan for the AO/GBT and Cassini datasets were inconsistent in Black et al.[2]. Furthermore, the Cassini dataset was internally inconsistent and some observations used an erroneous model[9]. We have corrected the locations of both datasets to the current model recommended by the International Astronomical Union[10] (Methods, Updated AO/GBT Subradar Locations on Titan and Supplementary Data 1). These corrections correspond to relative changes of 3–14 km, with an average change of 11 km. These changes are noteworthy because Titan's complex surface varies on these length scales.

A third distinction, is that Fig. 2 includes the complete Cassini radar instrument (called RADAR; ref. [11]) image dataset of Titan from all phases of the mission from 2004 to 2017. Compared to the dataset from the prime mission (2004–2008), this results in an ≈50% increase in the number of AO/GBT locations that were imaged by the RADAR instrument and also improves the time constraints on temporal changes. A fourth important distinction, is the addition of the Cassini RADAR altimetry observations (Methods, Cassini RADAR Altimetry Observations). The NRCS of Titan's surface depends sensitively on incidence angle (e.g., refs. [2,7,8]) and the AO/GBT ASRR are reflections from the surface at nadir (0°) incidence. The incidence angles of the Cassini RADAR images, of the AO/GBT locations, however, range from 10 to 77°. In contrast, Cassini RADAR altimetry observations have near-nadir incidences (deviations from which are due to

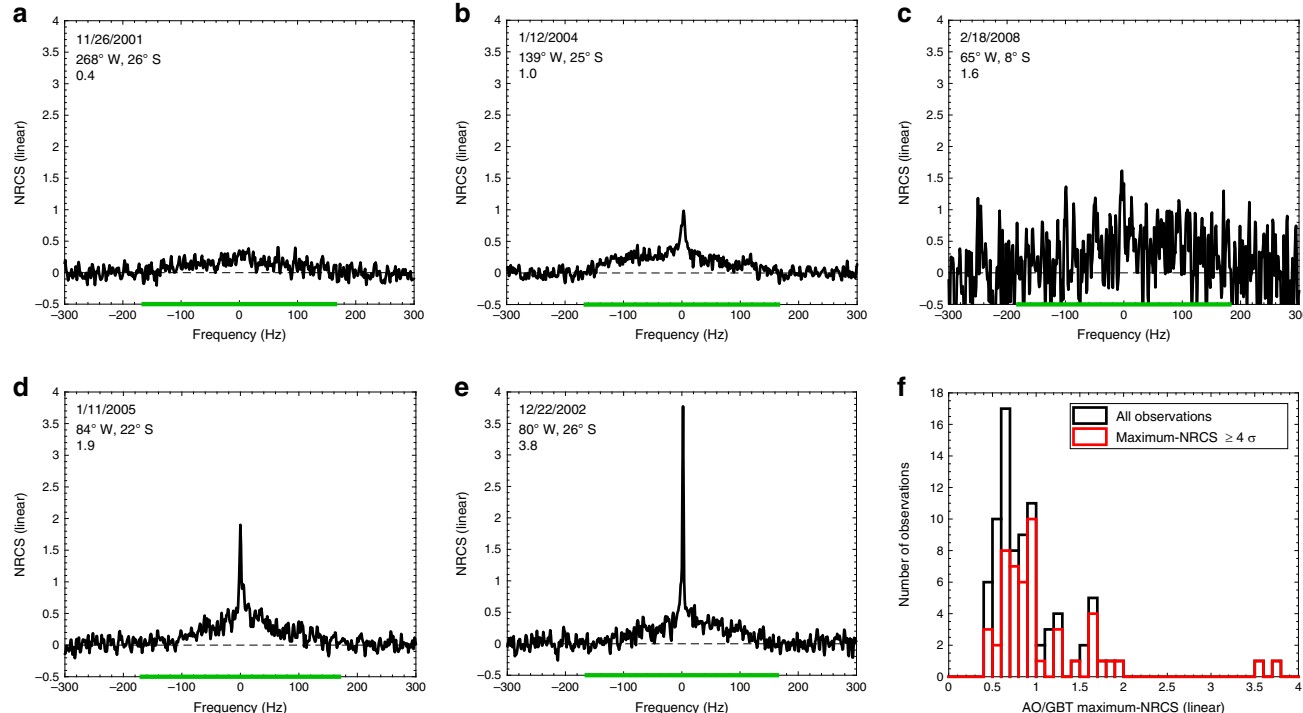

**Fig. 1 Arecibo Observatory and Green Bank Telescope observations of Saturn's moon Titan. a**–**e** Example AO/GBT echo spectra of Titan in the opposite circular polarization channel, adapted from Black et al.[2]. The observation date, subradar location on Titan, and maximum-NRCS are in the panel legends. The width of the echo from Titan is indicated by the green line on the x-axis; data beyond the green line are indicative of the noise. All AO/GBT spectra from Titan have a broad, diffuse component with an NRCS of approximately zero at the limits of the echo (endpoints of the green line on the x-axis) that increases toward the center. The NRCS of the central Doppler bin varies drastically between the spectra. Observations with anomalously high peaks (e.g., **d** and **e**) are the AO/GBT anomalously specular radar reflections. **c** Example demonstrating that the maximum-NRCS of some observations may be affected by noise. **f** Histograms of the maximum-NRCS of AO/GBT observations of Titan reported in Black et al.[2]. The black histogram includes all 83 observations and the red histogram includes only observations where the maximum-NRCS is ≥4 times the standard deviation of the noise ($\sigma$), to distinguish between observations with a high maximum-NRCS that may be due to greater noise and those that are confidently anomalous. Two observations have a maximum-NRCS ≥3.5 and are clearly separate from the distribution and thus anomalously specular. Another group of observations, with 1.5 ≈< maximum-NRCS ≤2, is also separate from most observations; these observations are also anomalously specular, albeit less so than the two observations with maximum-NRCS ≥3.5.

imperfect pointing). Thus, comparison of AO/GBT maximum-NRCS with Cassini-altimetry NRCS is a nadir-to-nadir comparison where the substantial effects of incidence angle are mitigated. This comparison, however, still has systematic differences since the AO/GBT observations include backscatter from non-nadir incidence angles along the height of the slit, and are at a radar wavelength ($\lambda$) of 12.6 cm, different from the Cassini RADAR ($\lambda = 2.2$ cm).

To distinguish between AO/GBT subradar locations with a high maximum-NRCS that may be due to greater noise and those that are confidently anomalous (e.g., Fig. 1a–e), locations where the maximum-NRCS was ≥4 times the standard deviation of the noise are colored red in Fig. 2. Thus, dots that are both red and large indicate locations that are confidently (i.e., low noise), anomalously specular (high maximum-NRCS) to AO/GBT. The large, red dots are concentrated at ≈70–135°W, 15–30°S. This region similarly has a relatively high NRCS in many Cassini altimetry observations (each altimetry track has many independent echoes).

The black areas in the polar regions of Fig. 2a are Titan's lakes/seas[4]. Their low NRCS and stark contrast with surrounding terrains is distinctive in Cassini RADAR images[4], and no such features were observed at any of the AO/GBT subradar locations, anomalously specular or non-specular[2]. Additional hypotheses for the AO/GBT ASRR include transient liquids, dunes/interdunes, and paleolakes/paleoseas.

**Transient liquids hypothesis.** Surface darkening following the observation of nearby clouds, most likely from ponding of methane rain, was observed twice by Cassini. The first rain event was in 2004 at ≈80°S (ref. [12]), far from the subradar locations of the AO/GBT observations, and the second in 2010 at ≈20°S (ref. [5]), which postdated all the AO/GBT observations. Thus, transient liquids from these two known rain events cannot explain the ASRR. The surface darkened by the 2010 tropical rain event did not completely revert to its original spectrum until >1 year after its initial darkening[13], demonstrating that surface changes from rain may be detectable long after the rainfall. Many other clouds were observed on Titan by both Cassini and Earth-based telescopes (e.g., refs. [14–17]), but they have not been associated with subsequent surface changes. Clouds over Titan's southern tropical region were uncommon during the period from 2004 to 2008 when both Cassini and AO/GBT observed Titan.

Nine AO/GBT locations were imaged by Cassini RADAR both before and after the AO/GBT observation; no surface changes are detected between the RADAR before and after images. Four RADAR images are <1 year from the AO/GBT observation and 13 images have an interval of <2 years; none of the images show surface liquids. Thus, the Cassini RADAR images do not support the hypothesis of transient liquids but also do not strongly constrain their presence during the AO/GBT observations. The low frequency of rain events on Titan (two

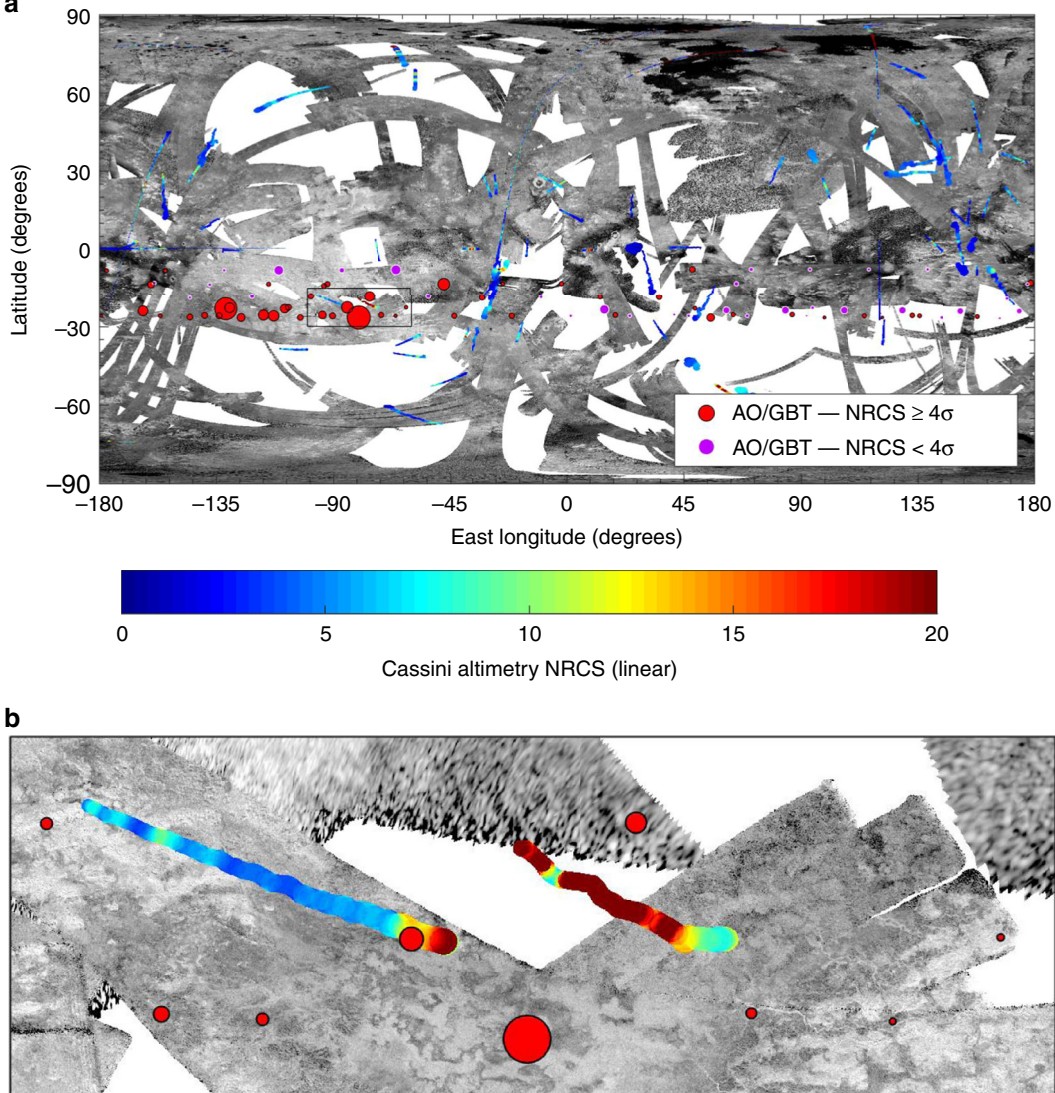

**Fig. 2 Map of Titan. a** The monochrome swaths are Cassini RADAR[11] images. The purple and red dots are AO/GBT subradar locations; red dots are locations where the maximum-NRCS was ≥4 standard deviations above the noise. Dot radii are linearly proportional to maximum-NRCS. Colored tracks are Cassini RADAR altimetry observations where color indicates NRCS. There is a high concentration of large, red dots at ≈70–135°W, 15–30°S and many Cassini altimetry observations in this region also have a high NRCS. **b** Boxed area enlarged. The colorbar applies to both **a** and **b**.

detected events over ≈13 years), long reversion timescale (>1 year[13]), low frequency of clouds over Titan's southern tropical region from 2004 to 2008 (refs. [15–17]), and RADAR image constraints all suggest that transient liquids are not responsible for the AO/GBT ASRR.

Black et al.[2] noted that the AO/GBT reflections depend on date and/or latitude but these variables are correlated in the AO/GBT observations and cannot be separated (see their Fig. 19). Comparison with Cassini altimetry in Fig. 3a–d indicates that latitude, not date, is responsible for the apparent dependence. Note that in Black et al.[2] there is a trend with date that is lost when the $y$-axis is changed from percentage of observations modeled with a specular component to maximum-NRCS. The trend with date is neither consistent with the choice of $y$-axis nor between AO/GBT and Cassini (Fig. 3b, d) and thus any temporal change hypothesis (including transient liquids) is unlikely to explain the AO/GBT ASRR. An explanation that does not invoke temporal changes is presented below. In summary, temporal changes are an unlikely hypothesis.

**Dunes/interdunes hypothesis**. Titan has dunes analogous to sand dunes on Earth except that the grains are likely composed of hydrocarbons rather than silicates (e.g., ref. [18]). Dune fields cover millions of square kilometers on Titan, primarily in its tropical region, and are one of Titan's main terrain units. Although the dune fields have rough surfaces at the scale of individual dunes (≈1 km), and possibly also at the scale of individual grains (≈1 mm), the surface of each dune and interdune may be smooth, as on the Earth, at the scale of the AO/GBT observation wavelength ($\lambda = 12.6$ cm, ref. [19]). The AO/GBT ASRR are from Titan's tropical region. From this cursory perspective, it is tempting to relate the ASRR to the dunes or interdune areas (which are brighter than the dunes in Cassini altimetry observations[19]). On closer inspection, however, there is no correlation. Seven AO/GBT subradar locations are clearly dune/interdune regions in Cassini RADAR images and their AO/GBT maximum-NRCS range from 0.56 to 1.09. From Fig. 1f, these maximum-NRCS values are not anomalous and do not include the observations of interest. Thus, the dune fields are ruled out as the source of the AO/GBT ASRR.

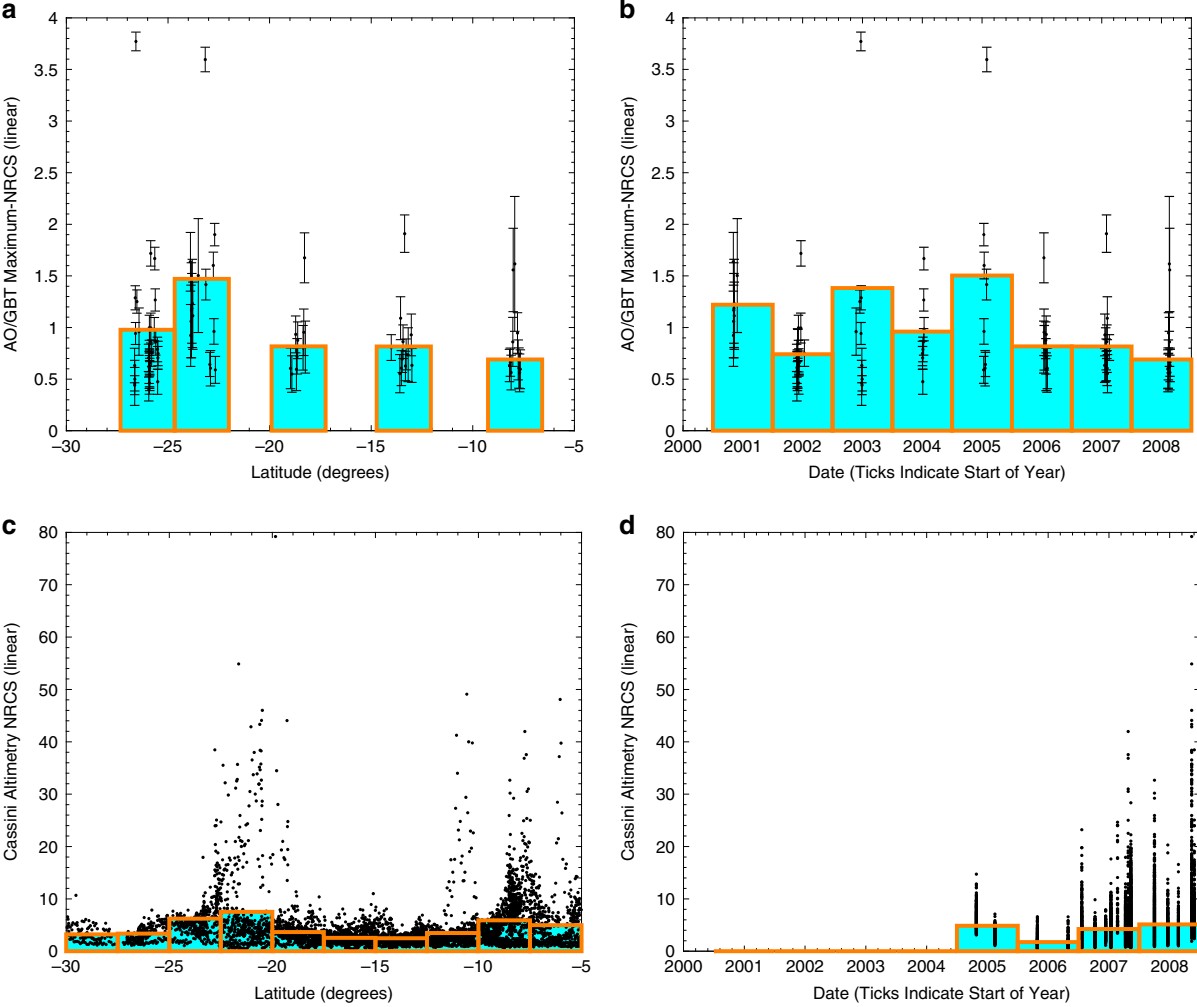

**Fig. 3 NRCS as a function of latitude and date for AO/GBT and Cassini altimetry observations.** Panels **a**, **b** show AO/GBT observations, like Fig. 19 of Black et al.[2] but the y-axis variable is maximum-NRCS. Panels **c**, **d** show Cassini altimetry observations. Panels **a**, **c** show latitude and panels **b**, **d** show date. Black dots are observations and orange-edged, cyan-filled rectangles are weighted means of the bins. Error bars show one standard deviation of the noise. There is an approximately consistent peak of the cyan/orange bins at ≈23°S between AO/GBT and Cassini and the latitude trends are somewhat similar. There is no agreement in the trends with date. Thus, the anomalously specular AO/GBT observations depend on location, not date.

**Hotei and Tui Regiones and paleolake/paleosea hypothesis.** The AO/GBT ASRR are concentrated at ≈70–135°W, 15–30°S (Fig. 2); the southeastern part of this area is Hotei Regio, Tui Regio is the southwestern part (boundaries shown in Fig. 4a), and the area in-between is southern Xanadu. Hotei and Tui Regiones have very similar properties and together are distinct terrains in Titan's tropical region in their infrared spectrum, morphology, topography, and as we show in this paper, nadir radar brightness. They have been hypothesized to be cryovolcanic terrains on the basis of their spectroscopy, morphology, and topography (e.g., refs. [20–22]). On the other hand, they have been hypothesized to be paleoseas/paleolakes, also based on their spectroscopy[23,24] and morphology and topography[25]. Spectrally, their most distinct signature is an unusually high brightness in Titan's 5-μm atmospheric window. MacKenzie et al.[24] mapped Titan's 5-μm-bright areas using Cassini Visual and Infrared Mapping Spectrometer (VIMS)[26] observations and we use their map as the definition of the extent of Hotei and Tui Regiones in this work. The map is shown in Fig. 4a with the AO/GBT observations as in Fig. 2.

Figure 4a shows that every AO/GBT observation with a subradar point in Hotei/Tui is anomalously specular, including the two observations that are distinctly brighter than all others

(the two largest, red dots; one is in Hotei Regio and the other Tui Regio). Of the two highest (maximum-NRCS ≥3.5) plus seven high confidence (red), intermediate maximum-NRCS (1.5 ≤ maximum-NRCS ≤ 2, Fig. 1f) observations, six are in Hotei or Tui Regiones, one is on Hotei Regio's boundary, and only two are not in or near either Regio. The two exceptions merit an extended discussion, which is deferred to the next section. Regardless, we conclude that Hotei and Tui Regiones are the primary source of the AO/GBT ASRR.

The subradar locations of the two distinctly specular AO/GBT observations were imaged by Cassini RADAR. Figure 4b, d show that the AO/GBT subradar tracks (the subradar point moves as the echo is integrated over time, primarily from Titan's rotation) are over bright areas. These bright areas are morphologically similar to paleolakes (called empty lake basins[4]) in Titan's polar regions[25]. Figure 4c shows that the bright areas in Hotei Regio are topographic lows, consistent with the paleolake hypothesis (stereo topography does not cover the analogous location in Tui Regio). Hotei Regio also has a relatively high NRCS in many Cassini altimetry observations (Fig. 2; Tui Regio was not observed with Cassini altimetry). However, the highest-NRCS Cassini altimetry observations are of Titan's liquid-filled lakes/seas: coherent

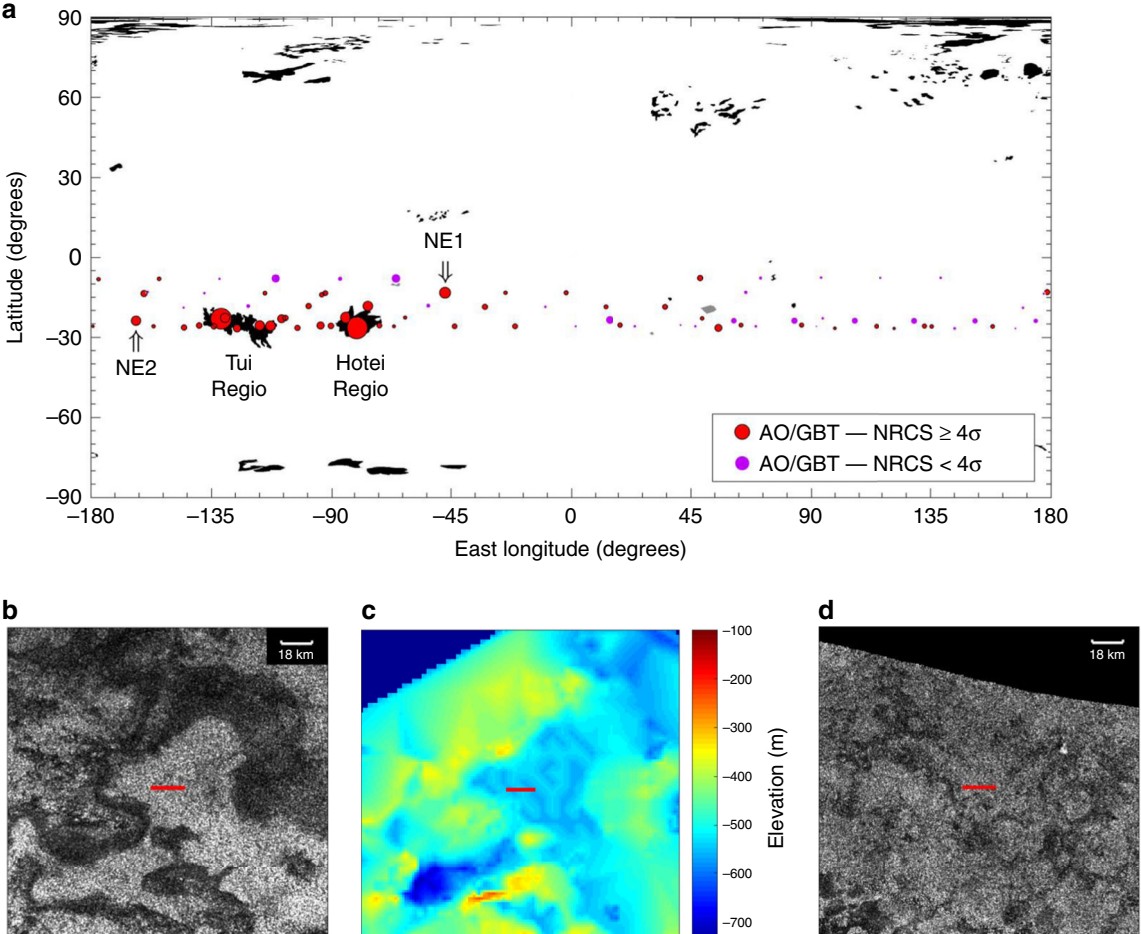

**Fig. 4 Hotei and Tui Regiones. a** Black areas are Titan's 5-µm-bright regions as mapped in MacKenzie et al.[24]. Hotei and Tui Regiones are the 5-µm-bright regions at ≈80°W, 25°S and ≈120°W, 25°S. AO/GBT observations are plotted as in Fig. 2. Hotei and Tui Regiones are the main source of the AO/GBT anomalously specular reflections. Two noteworthy exceptions to the strong correlation of anomalously specular AO/GBT observations to Hotei/Tui are labeled NE1 and NE2 and discussed in the Discussion section. **b** The 18-km subradar track (red line) of the highest maximum-NRCS (and most specular in Black et al.[2]) AO/GBT observation, located in Hotei Regio (largest red dot in Hotei Regio in **a**). **c** The stereo topography of the same area[37]. The AO/GBT reflection is from a bright area that is morphologically similar to paleolakes[25] (**b**) and is topographically low (**c**). **d** The 18-km subradar track (red line) of the second highest maximum-NRCS (and second most specular in Black et al.[2]) AO/GBT observation, located in Tui Regio (largest red dot in Tui Regio in **a**). The AO/GBT reflection is again from a bright area that is morphologically similar to the comparably specular location in Hotei Regio and paleolakes. Stereo topography does not cover this track.

specular reflections from the lakes/seas can be 100 times brighter than the diffuse echoes from surrounding solid surfaces[27,28]. Thus, in Cassini altimetry observations, Hotei Regio is not as bright as the liquid-filled lakes/seas but is brighter than many other terrains. Hotei Regio has a higher NRCS than other tropical regions in both AO/GBT and Cassini altimetry observations; it is bright in nadir radar observations. The paleolakes in Titan's north polar region are similarly brighter than their surroundings in Cassini altimetry observations but not as bright as the liquid-filled lakes/seas[29,30]. Consequently, we add Cassini-altimetry-brightness to the list of characteristics that support the paleolake/paleosea hypothesis for Hotei Regio and, by analogy, Tui Regio. We favor this hypothesis but note that the cryovolcanic and paleolake/paleosea hypotheses are not mutually exclusive. We also note that our earlier conclusion that Hotei/Tui are the primary source of the AO/GBT ASRR is independent of the interpretation of their geomorphology; regardless of whether they are paleolakes/paleoseas, cryovolcanoes, or another terrain. Based on the above correlations we predict that: (1) the bright regions of Tui Regio in

RADAR images are topographic lows and (2) Tui Regio will have an intermediate NRCS in RADAR nadir observations.

The smoother a surface, the greater its NRCS in nadir radar observations. The high NRCS of Hotei/Tui in AO/GBT and Cassini altimetry observations, however, cannot be attributed solely to the smoothness of their surface. As shown in Fig. 4b, d, the subradar locations of the anomalous AO/GBT observations are also bright in Cassini RADAR images, which have nonzero incidence angles (the incidence angles for Fig. 4b, d were 18° and 23°). A greater dielectric constant and greater diffuse scattering both increase the NRCS at all incidence angles. The dielectric constant of a surface depends on its composition (and other parameters, e.g., porosity) and Hotei/Tui have a different composition than their surroundings; recall that they are spectrally distinct, particularly in Titan's 5-µm atmospheric window (e.g., ref. [24]). Thus, the surfaces of Hotei/Tui likely differ from other surfaces in Titan's tropical region in both their smoothness and dielectric constant (composition) and both of these differences contribute to their high maximum-NRCS in

AO/GBT observations. These differences may be due to lake/sea sediments that were deposited in the past, when Hotei/Tui were liquid-filled[31,32]. Further study of Hotei/Tui may be fertile research for understanding the geology and chemistry of Titan's lake/sea sediments and the long-term evolution of its climate.

## Discussion

There are two noteworthy exceptions to the strong correlation of AO/GBT ASRR to Hotei/Tui. The first has a subradar point at ≈47°W, 13°S (NE1 in Fig. 4a), was acquired on 1/30/2007, and has the third-highest maximum-NRCS of all the AO/GBT observations. Based on the conclusions above, that location is predicted to be morphologically similar to Hotei/Tui/paleolakes, topographically low, and bright in 5-μm and nadir radar observations. That location is at the edge of two low-resolution Cassini RADAR images but noise inhibits interpretation of the surface. It was observed by VIMS and intriguingly is at an intersection of the equatorial-bright (organic sediments) and dark-blue (water ice enriched) spectral units, not the 5-μm-bright (evaporite) spectral unit (e.g., refs. [33,34]). The second exception has a subradar point at ≈163°W, 24°S (NE2 in Fig. 4a) and was acquired from 11/3/2000. We have dismissed other bright reflections centered on the 2001 and 2008 bins in Fig. 3 based on their greater noise (primarily a result of shorter integration times in the first and last oppositions of the AO/GBT campaign) but this observation was almost six standard deviations above the noise and cannot be dismissed. Nevertheless, we have some reservation about predicting the terrain at that location is similar to Hotei/Tui. This was the first observation of the AO/GBT campaign, the maximum NRCS is not from the central Doppler bin, and the spectral spike is only one bin wide. This location was not observed with RADAR but was observed by VIMS and its spectrum is a mix of the equatorial-bright and 5-μm-bright spectral units. The Cassini mission ended in 2017 so these locations cannot be imaged in high-resolution until another spacecraft explores Titan.

The nadir-to-nadir comparison between AO/GBT and Cassini altimetry observations also leads to an important conclusion about the definition of specular in the context of liquids on a planetary surface. The AO/GBT observations of Hotei and Tui Regiones are specular in several senses: they have a spike in the central Doppler bin of their opposite circular polarization spectrum, they have a high NRCS, and they are best fit by radar scattering models that include a specular component. The Cassini altimetry observations of Hotei Regio also have a high NRCS, supporting the notion that Hotei and Tui Regiones are specular terrains. However, Cassini altimetry observations of Titan's lakes/seas indicate that reflections from Titan's liquid surfaces are markedly more specular (brighter). The reflections from the smooth lake/sea surfaces are so bright that the echoes must be coherent[27,28]. The received power in this case decreases with the square of the total (two-way) distance from transmitter to reflector to receiver ($(2d)^2$ for monostatic observations, where $d$ is distance from transmitter/receiver to reflector) rather than the square of the distance from transmitter to reflector multiplied by the square of the distance from reflector to receiver ($d^4$ for monostatic observations) as it does for noncoherent reflections. Using Titan as a solar system ground-truth in the search for oceans on exoearths suggests that, to identify liquids by specular reflections, a stringent definition of specular should be used. We recommend a definition based on the coherence of the reflected electromagnetic waves rather than definitions based on combinations of relative brightness, incidence angle, location, and/or polarization; reflections that are specular in the latter senses are quasi-specular. For Titan, liquids were expected based on the planetary context and reflections that are specular in the relative brightness, incidence angle, expected location, and polarization senses were observed but the reflections were from solid, not liquid, surfaces. The smooth liquid surfaces on Titan, however, are specular in a coherent sense at microwave[27,28] and infrared (e.g., refs. [35,36]) wavelengths.

## Methods

**Updated AO/GBT subradar locations on Titan**. The AO/GBT subradar locations on Titan were updated from Black et al.[2] using the International Astronomical Union's (IAU) current (2015) spin model[10]. The start and end times of each AO/GBT observation (provided in Supplementary Data 1) were input into the Navigation and Ancillary Information Facility's (NAIF) SPICE system. The subradar points on Titan, at the time the radar echo departed from Titan's surface, were calculated using the cspice_subpnt module. The Titan ellipsoid was a sphere with a radius of 2575 km. The updated locations are provided in Supplementary Data 1. Note that the Cassini RADAR images and altimetry observations were also located using the same spin model and ellipsoid as part of the pipeline processing of the mission datasets.

**Cassini RADAR altimetry observations**. In both Figs. 2 and 3, only Cassini RADAR altimetry observations with an incidence angle of <1° are included. The cut removes echoes acquired with substantial pointing errors as well as echoes at the beginning and end of the tracks. We verified that changing the cutoff does not qualitatively affect the results. The NRCS of each altimetry echo was corrected for deviations from nadir (0° incidence) pointing; the incidence angle correction is described in the Cassini RADAR Users Guide, which is available from the NASA Planetary Data System (PDS). We found that varying the incidence angle correction, even not including a correction, does not qualitatively affect the results.

## Data availability

The complete AO/GBT dataset is thoroughly described and presented in Black et al.[2]. The updated AO/GBT subradar locations are provided in Supplementary Data 1. The complete Cassini RADAR dataset, including all images, altimetry observations, and stereo topography is available from the NASA Planetary Data System (PDS) at: https://pds-imaging.jpl.nasa.gov/volumes/radar.html. Titan Trek is a NASA web-based portal with a browsing tool that allows for easy viewing and layering of Cassini RADAR images as well as other Cassini data of Titan; it is available at https://trek.nasa.gov/titan/.

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

## Acknowledgements

We are sincerely grateful to the entire Cassini, Arecibo Observatory, and Green Bank Telescope teams for enabling this research. J.D.H. gratefully acknowledges the Natural Sciences and Engineering Research Council of Canada, Post Graduate Scholarship Program and NASA Postdoctoral Program for financial support. J.D.H. thanks Bonnie Buratti for professional mentorship. Part of this research was carried out at the Jet Propulsion Laboratory, California Institute of Technology, under a contract with the National Aeronautics and Space Administration. R.D.L. acknowledges the support of NASA Grant 80NSSC18K1389.

## Author contributions

J.D.H. led the analysis and writing of the article. A.G.H., D.B.C., and J.I.L. worked closely with J.D.H. on all aspects of the analysis and writing. G.J.B., S.M.M., and R.D.K. assisted with the AO/GBT, VIMS, and stereo topography analyses respectively. All authors contributed to the data acquisition and discussions.

## Competing interests

The authors declare no competing interests.
