## [Peer Review File · Nature Communications]

Reviewers' comments:

Reviewer #1 (Remarks to the Author):

The manuscript discusses the discrepancies between the specular reflections detected by the Arecibo Observatory/Green Bank Telescope and the instruments onboard the Cassini spacecraft. The detection of specular reflections are an important tool in order to identify recently present liquids on Titan's surface.

In this work, however, the authors present new evidence that the Arecibo data rather point to paleolakes and paleoseas than recently existing liquids. Of particular interest are the areas of Tui and Hotei Regio. Several former studies have proposed that these surface features are cryo-volcanic flows or possibly present dry lakes/seas. Thus this work provides important evidence to reveal the nature of these key features on Titan. It also expands the usage of specular reflections i.e. the NRCS not only for the detection of recent liquids on a planetary surface but also areas, which have been covered by liquids in the past. Thus this work significantly furthers or knowledge of Titan's methane hydrologic cycle. This work will be certainly be of high interest to the scientific community, in particular with the increasing interest in the study of exoplanets. Thus, I gladly recommend the publication of this work. I only have a few small points, which the authors might consider.

1. The paper jumps from the abstract to the results. I rather would use the second paragraph of the abstract as a short general introduction into the topic and motivation of the presented work.
2. I recommend to remove the VIMS data from Figure 1. They are difficult to recognize and considerably decrease the contrast of the radar images. It is enough to show the VIMS 5- μm bright spots in Figure 3. Instead I would consider to show the subsets VIMS 5 μm data of Tui and Hotei Regio similar to Figs. 3b and d. This would be particularly interesting for Tui regio, where no stereo topography is available.

Reviewer #2

Report on manuscript NCOMMS-19-15755:

"The Root of Specular Reflections from Solid Surfaces on Saturn's Moon Titan"

by Hofgartner, Hayes, Campbell, Lunine, Black, MacKenzie, Birch, Elachi, Kirk, Le Gall, Lorenz, Wall

This paper is interesting and proposes an interpretation of Arecibo and Green Bank radar specular observations of some low latitude regions. The authors propose that the mentioned observations are correlated to possible paleolakes/paleoseas discussed by MacKenzie *et al.*, on the basis of infrared spectro-imaging. However, some aspect (detailed below) have to be clarified. For instance, the "anomalous reflections" have to be identified quickly in the text ... don't forget that *Nature Communication* has a pretty broad audience. Similarly, it is not, for me, always clear what we observed: distribution of incident angle? ... ?

Beginning of the text: the authors should be very clear concerning the "anomalous reflections". Are they the two around NRCS ~ 3.5-4.0 acquired in 2003 and 2005?

I. 55-99 and Fig. 1: How did the authors deal with the georeferencing with Cassini and ensure consistency with AO/GBT observations? Did they make estimations of uncertainties related to this technical aspect?

My feeling is that AO/GBT geometry could be much less constrained than Cassini ones.

I. 88-91: "The NRCS of Titan's surface depends sensitively on incidence angle (e.g., ref. 2,8,9) and the AO/GBT specular reflections are reflections from the surface at nadir (0°) incidence. The incidence angles of the RADAR images of the AO/GBT locations, however, range from 10-77°"

> So, are 0° incidence observations included in AO/GBT set? If the letter range between 10 and 77° this is not the case. Please, be more clear.

Fig 1 caption: "Figure 1: A. Map of Saturn's moon Titan. The monochrome swaths are Cassini RADAR 7 images, which are overlain on a Cassini VIMS 34 mosaic. The purple and red dots are AO/GBT subradar locations; red dots are locations where the maximum-NRCS was ≥ 4 standard deviations above the noise. Dot radii are linearly proportional to NRCS. Colored tracks are Cassini RADAR altimetry observations where color indicates NRCS. There is a high concentration of large, red dots at $\approx 70-135^\circ$ W, $15-30^\circ$ S and many Cassini altimetry observations in this region also have a high NRCS. B. Boxed area enlarged."

> in Fig 1, many Cassini radar swaths indicates a NRCS not really correlated to AO/GBT NRCS (or even frankly anticorrelated), for instance :

How can the authors explain this aspect?

> In both cases (AO/GBT and Cassini): specular nadir reflections?

l. 118-121: “The southeastern part of this area is Hotei Regio, Tui Regio is the southwestern part, and the area in-between is southern Xanadu. Hotei and Tui Regiones have very similar properties (see below). Figure 1 shows that Hotei Regio has a high NRCS in both AO/GBT and Cassini altimetry observations.”

> to improve the readability please indicate Hotei Regio, Tui Regio and Xanadu on the map of Fig. 1, for instance using contours (with clear labels like “Hotei”, “Tui” and “Xanadu”).

l. 139-140: “Nine AO/GBT locations were imaged by RADAR before and after the AO/GBT observation; no surface changes are detected.”

> Not logical, please reformulate your sentence ... “... before and after the AO/GBT **specular (?)** observation...”

l. 143-147:

"The low frequency of rain events on Titan, long reversion timescale, low frequency of clouds over Hotei and Tui Regiones, and RADAR image constraints all suggest that transient liquids are not responsible for the AO/GBT specular observations. An explanation that does not invoke temporal changes is presented later in this paper. Transient liquids are therefore an unlikely and unnecessary hypothesis."

> A few references are needed here. The point is important and need to be supported by some additional theoretical/observational arguments. Even if they are far to be perfect (particularly concerning the representation of the methane cycle) don't we have any clue in term of rainfalls frequency from GCM? On the VIMS/ISS side: are there new observations analysis showing some indication of rainfalls frequency? Rodriguez et al.? Corlies et al.? I would be more easily convinced by an (even crude) estimation of the **probability** of observing a transient liquid feature. No way to be a little bit more quantitative?

But, I recongize that the authors are probably right by stating: “Transient liquids are therefore an unlikely and unnecessary hypothesis.”

Fig. 2: All the data collected in these plots are nadir observation? No distribution of incident angles (to be sure we are observing a distribution of incident angles)? Please be clear. Please show (with another colour?) observations which are considered “anomalous”. The “peak” observed in Cassini data is not really in accordance with AO/GBT data since for Cassini with have a peak position around $\sim 23^\circ$, while for AO/GBT we get a position higher by a few degrees, please put ticks every degree.

l. 170-185:

The author clearly identify the “anomalous reflections” only at this point of the text ... in my opinion they should be identify before.

> In Fig. 2 please put anomalous reflections measurements in another color.

l. 87-196:

“Titan has dunes analogous to sand dunes on Earth except that the grains are likely composed of hydrocarbons (tholins) rather than silicates (e.g., ref. 15). Dune fields cover millions of square kilometers on Titan, primarily in its tropical region, and are one of Titan’s main terrain units. The AO/GBT specular reflections are from Titan’s tropical region. From this cursory perspective, it is tempting to relate the specular reflections to the dunes or interdune areas (which are brighter than the dunes in Cassini altimetry observations 16). On closer inspection, however, there is no correlation. Seven AO/GBT locations are clearly dune/interdune regions in RADAR images and their AO/GBT maximum-NRCS range from 0.56-1.09. From Figure 2, these NRCS values are not anomalous and do not include the specular reflections of interest. Thus, the dune fields are decidedly ruled-out as the source of the AO/GBT specular reflections.”

> Very well, but why dune fields should be considered as very efficient reflectors? Since they are pretty rough ... a priori they should be bad reflector! What is the underlying physical process/scenario assumed by the authors?

Fig. 3: is the most convincing argument up to this point. The correlation between AO/GBT reflection and 5 microns bright zones identified by MacKenzie *et al.* is striking.

l. 226-227: How the radar brightness of these possible paleolakes/paleoseas compares with the radar brightness of river dry beds proposed by Le Gall *et al.* (2010)?

l. 254: “The dielectric constant of a surface depends on its composition (and other parameters) ... ”: what are these “other parameters”? Chemical composition is not the main parameter?

l. 266-267: “There are two noteworthy exceptions to the strong correlation of specular AO/GBT observations to Hotei/Tui. The first has a subradar point at $\approx 47^\circ\text{W}$, 13°S , ... “ : please show this point with a label, an arrow ... in one map.

l. 275-275: “The second exception has a subradar point at $\approx 163^\circ\text{W}$, 24°S and was acquired on 11/3/2000.” : idem, label in a figure/map.

l. 299-304: “Using Titan as a solar system ground-truth in the search for oceans on exoearths suggests that, to identify liquids by specular reflections, a stringent definition of specular should be used. We recommend a definition based on the coherence of the reflected electromagnetic waves rather than definitions based on combinations of relative brightness, incidence angle, location, and/or polarization; reflections that are specular in the latter senses are quasi-specular.”

> For “exo-ocean-planet”, the integration time of an electromagnetic signal could be quite long compared to timescale associated to waves, this could significantly alter the coherence of the signal. But, the author are right encouraging scientific discussions about what could be the best criterion for the identification of specular reflection.

Reviewer #3 (Remarks to the Author):

I do think the results presented in this manuscript would be of interest to the planetary community. However, to improve the impact factor, I recommend a rewrite of the manuscript to clean up some problems. The main issue is the way the manuscript is written – it is highly disconnected to a point that it is difficult to read. Grammatically, the manuscript is fine. However, due to the disjointed nature of the writing, the manuscript is very hard to follow and therefore not as easy to interpret. For example, consider the sentence starting with “Thick haze and abundant methane...” on lines 31

and 32. The sentence is out of place and has nothing to do with the sentences around it. My guess would be that the author is trying to tell a story about the aerosol, which obscures Titan’s lower atmosphere and surface at visible wavelengths – this is justification for going to longer wavelengths. But, this is well known and as written, it is highly disconnected and out of place, which is true for much of the manuscript. It almost reads like random thoughts were placed together, which borders on being incoherent. This type of disconnect in the writing appears throughout the entire manuscript.

The structure of the manuscript is also rather odd. It begins with a two paragraph Abstract, followed by a very long Result section with no subsections, and ends with a Discussion section. As written, I’m not convinced that ground-based AO/GBT observations are indeed specular. This impact factor is simply not in the as-written manuscript. A major rewrite is needed especially if published in a higher quality journal.

In the Results section, when you call out Fig. 1, make sure to state in the text (just like you do in the figure caption) that these are maps from the Cassini RADAR instrument. Stating just a “Cassini map” is careless. Also, when were the RADAR swaths acquired? I would add the time and Cassini flybys in the text and captions, and also do the same for the VIMS data shown.

If you are going to dive into four very long explanations as to how Figure 1 is different than Figure 17 in Black et al., then you need to first introduce what Black et al. was reporting. Instead, I’d add a subsection and discuss how this work effort compares to other efforts. And, make sure to show more figures, especially talk to the ones you keep referring to in other publications (like those of Black et al.).

To improve the manuscript, I would really rethink the figures. Fig. 1B doesn’t add much. Fig.

2 B and D don't add much either – A and C carry the impact factor. Do you really need Fig. 3A since you have Fig. 1A? Fig. 3B,C, and D are really good.

Reviewer #1

Thank you for this positive and helpful review, it has improved the manuscript.

The manuscript discusses the discrepancies between the specular reflections detected by the Arecibo Observatory/Green Bank Telescope and the instruments onboard the Cassini spacecraft. The detection of specular reflections are an important tool in order to identify recently present liquids on Titan's surface. In this work, however, the authors present new evidence that the Arecibo data rather point to paleolakes and paleoseas than recently existing liquids. Of particular interest are the areas of Tui and Hotei Regio. Several former studies have proposed that these surface features are cryo-volcanic flows or possibly present dry lakes/seas. Thus this work provides important evidence to reveal the nature of these key features on Titan. It also expands the usage of specular reflections i.e. the NRCS not only for the detection of recent liquids on a planetary surface but also areas, which have been covered by liquids in the past. Thus this work significantly furthers or knowledge of Titan's methane hydrologic cycle. This work will be certainly be of high interest to the scientific community, in particular with the increasing interest in the study of exoplanets. Thus, I gladly recommend the publication of this work. I only have a few small points, which the authors might consider.

1. The paper jumps from the abstract to the results. I rather would use the second paragraph of the abstract as a short general introduction into the topic and motivation of the presented work.

We added the section title "Introduction" to the introduction section, which separates the abstract and results section. We also expanded this section.

2. I recommend to remove the VIMS data from Figure 1. They are difficult to recognize and considerably decrease the contrast of the radar images. It is enough to show the VIMS 5- μm bright spots in Figure 3. Instead I would consider to show the subsets VIMS 5 μm data of Tui and Hotei Regio similar to Figs. 3b and d. This would be particularly interesting for Tui regio, where no stereo topography is available.

We removed the VIMS data from figure 1 and increased the contrast of the RADAR images.

The VIMS observations have significantly lower spatial resolution than the Cassini SAR observations so, unfortunately, panels similar to figure 3B and 3D are somewhat pixelated and not particularly informative.

Reviewer #2

Thank you for these detailed comments, they have improved the manuscript.

This paper is interesting and proposes an interpretation of Arecibo and Green Bank radar specular observations of some low latitude regions. The authors propose that the mentioned observations are correlated to possible paleolakes/paleoseas discussed by MacKenzie et al., on the basis of infrared spectro-imaging. However, some aspect (detailed below) have to be clarified. For instance, the “anomalous reflections” have to be identified quickly in the text ... don't forget that Nature Communication has a pretty broad audience. Similarly, it is not, for me, always clear what we observed: distribution of incident angle? ... ?

We added several paragraphs and a figure to the start of the results section to quickly and clearly introduce the anomalous reflections.

The added material at the start of the results section also explains the AO/GBT observations in more detail.

Beginning of the text: the authors should be very clear concerning the "anomalous reflections". Are they the two around NRCS ~ 3.5-4.0 acquired in 2003 and 2005?

We added several paragraphs and a figure to the start of the results section to quickly and clearly introduce the anomalous reflections.

I. 55-99 and Fig. 1: How did the authors deal with the georeferencing with Cassini and ensure consistency with AO/GBT observations? Did they make estimations of uncertainties related to this technical aspect?

My feeling is that AO/GBT geometry could be much less constrained than Cassini ones.

The Cassini observations were georeferenced as part of the Cassini project's pipeline processing of the spacecraft data. The details are provided in the data users guide on the data archive, NASA's Planetary Data System, and in Stiles et al., 2010 (cited in the manuscript).

We ensured consistency by using the International Astronomical Union's (IAU) current (2015) model for Titan to georeference both the AO/GBT and Cassini observations. Our careful attention to defining the locations led to the discovery of inconsistencies in prior work, which we have corrected. This is discussed in point #2 of the Fig. 1 (now Fig. 2) distinctions from Black et al., 2011. We included the updated AO/GBT subradar locations in the supplementary material. We added a paragraph to the Methods section explaining how the AO/GBT subradar locations were updated and that the Cassini georeferencing is part of the pipeline processing.

The AO/GBT georeferencing is set primarily by the positions and spins of Titan and Earth and it is not as sensitive to telescope pointing errors as many other remote sensing observations. The timing of ground-based radar observations is always recorded to high-precision and with the

time it is possible to update the location of the observations using improved models for the positions and spins of Titan and Earth. Thus, the AO/GBT subradar locations are constrained to an accuracy comparable to the Cassini observations.

I. 88-91: "The NRCS of Titan's surface depends sensitively on incidence angle (e.g., ref. 2,8,9) and the AO/GBT specular reflections are reflections from the surface at nadir (0°) incidence. The incidence angles of the RADAR images of the AO/GBT locations, however, range from $10-77^\circ$ "

> So, are 0° incidence observations included in AO/GBT set? If the letter range between 10 and 77° this is not the case. Please, be more clear.

Yes, the central Doppler bin of each AO/GBT observation corresponds to a slit of Titan's disk that includes the 0 degrees incidence reflection. We expanded the explanation of the AO/GBT observations at the start of the results section so that the details of those observations should now be clearer to the reader.

Recall that RADAR, with capitals, is the Cassini radar instrument and the images from this instrument range from $10-77$ degrees. Changed sentence to: "The incidence angles of the Cassini RADAR images, of the AO/GBT locations, however, range from $10-77^\circ$."

Fig 1 caption: "Figure 1: A. Map of Saturn's moon Titan. The monochrome swaths are Cassini RADAR 7 images, which are overlain on a Cassini VIMS 34 mosaic. The purple and red dots are AO/GBT subradar locations; red dots are locations where the maximum-NRCS was ≥ 4 standard deviations above the noise. Dot radii are linearly proportional to NRCS. Colored tracks are Cassini RADAR altimetry observations where color indicates NRCS. There is a high concentration of large, red dots at $\approx 70-135^\circ$ W, $15-30^\circ$ S and many Cassini altimetry observations in this region also have a high NRCS. B. Boxed area enlarged."

> in Fig 1, many Cassini radar swaths indicates a NRCS not really correlated to AO/GBT NRCS (or even frankly anticorrelated), for instance :

How can the authors explain this aspect?

In the figure, the distinction between red and purple dots is related to the uncertainty of the NRCS measurements, not the magnitude of the NRCS. The size of the dots indicates the magnitude of the NRCS. The examples given above are all relatively small dots so have a

relatively low AO/GBT NRCS. In this case, the Cassini altimetry would also be expected to have a low NRCS and thus be blue. Thus, these are not examples of anticorrelation.

> In both cases (AO/GBT and Cassini): specular nadir reflections?

The AO/GBT and Cassini altimetry are both nadir observations. In the above examples, neither the AO/GBT nor the Cassini observations are very specular.

I. 118-121: "The southeastern part of this area is Hotei Regio, Tui Regio is the southwestern part, and the area in-between is southern Xanadu. Hotei and Tui Regiones have very similar properties (see below). Figure 1 shows that Hotei Regio has a high NRCS in both AO/GBT and Cassini altimetry observations."

> to improve the readability please indicate Hotei Regio, Tui Regio and Xanadu on the map of Fig. 1, for instance using contours (with clear labels like "Hotei", "Tui" and "Xanadu").

We moved these sentences to the subsection discussing Hotei and Tui Regiones. We also modified them to include pointers to Fig. 4, which does indicate the boundaries. These changes also improve the flow of the manuscript as Hotei and Tui Regiones are discussed in detail the first time they are mentioned.

I. 139-140: "Nine AO/GBT locations were imaged by RADAR before and after the AO/GBT observation; no surface changes are detected."

> Not logical, please reformulate your sentence ... "... before and after the AO/GBT specular (?) observation..."

Changed to: "Nine AO/GBT locations were imaged by Cassini RADAR both before and after the AO/GBT observation; no surface changes are detected between the RADAR before and after images."

I. 143-147:

"The low frequency of rain events on Titan, long reversion timescale, low frequency of clouds over Hotei and Tui Regiones, and RADAR image constraints all suggest that transient liquids are not responsible for the AO/GBT specular observations. An explanation that does not invoke temporal changes is presented later in this paper. Transient liquids are therefore an unlikely and unnecessary hypothesis."

> A few references are needed here. The point is important and need to be supported by some additional theoretical/observational arguments. Even if they are far to be perfect (particularly concerning the representation of the methane cycle) don't we have any clue in term of rainfalls frequency from GCM? On the VIMS/ISS side: are there new observations analysis showing some

indication of rainfalls frequency? Rodriguez et al.? Corlies et al.? I would be more easily convinced by an (even crude) estimation of the probability of observing a transient liquid feature. No way to be a little bit more quantitative?

But, I recongize that the authors are probably right by stating: "Transient liquids are therefore an unlikely and unnecessary hypothesis."

Changed to: "The low frequency of rain events on Titan (two detected events over ≈ 13 years), long reversion timescale (> 1 year, ref. ¹²), low frequency of clouds over Titan's southern tropical region from 2004-2008 (ref. ^{13,14}), and RADAR image constraints all suggest that transient liquids are not responsible for the AO/GBT ASRR."

The most-current, published results of cloud frequency from ISS/VIMS is Turtle et al., 2018, which is included as reference number 14 in the manuscript.

We agree that it would be nice to quantitatively estimate the probability of a rain event. However, the available data do not support even a crude estimation. Only two rain events were observed over approximately 13 years and neither was near the location of the anomalously specular AO/GBT observations. This suggests a very low (by terrestrial standards) frequency (~ 2 per 13 years if the location of any given rain event is random, < 2 per 13 years if the observed rain events occurred in Titan's high rain-frequency locations). However, the frequency could vary substantially with season (as it does on Earth) and Cassini could have missed rain events due to spatial resolution and/or temporal coverage. A quantitative guess, will all of these caveats, will likely distract the reader from the more important points that the rain frequency is low and that there is no good evidence to expect rain events to be responsible for the anomalous AO/GBT observations. We are also not aware of a reliable GCM prediction for this specific question.

Fig. 2: All the data collected in these plots are nadir observation? No distribution of incident angles (to be sure we are observing a distribution of incident angles)? Please be clear. Please show (with another colour?) observations which are considered "anomalous". The "peak" observed in Cassini data is not really in accordance with AO/GBT data since for Cassini with have a peak position around $\sim 23^\circ$, while for AO/GBT we get a position higher by a few degrees, please put ticks every degree.

We added several paragraphs and a figure to the start of the results section to quickly and clearly introduce the anomalous reflections. Since the anomalous observations are now identified well before this figure, we do not think it is necessary to highlight them here. We considered changing the color of the anomalous observations in this figure but found that it distracts from the main point of the figure (latitudinal vs. temporal dependence). A new figure, earlier in the results section, has been added whose main point is identifying the anomalous observations.

We agree that the peaks are not at exactly the same latitude. Note that we intentionally aligned the two figures vertically so that the reader could easily compare them and see the

similarities and differences. We consider this difference to be a result of the different latitudinal distributions of the two (AO/GBT and Cassini) observations over Hotei/Tui. Had the AO/GBT and Cassini observations covered exactly the same locations, we expect that they would have peaks at the same latitudes. We changed the sentence in the caption to: "There is an approximately consistent peak of the cyan/orange bins at $\approx 23^\circ\text{S}$ between AO/GBT and Cassini and the latitude trends are somewhat similar." The main point is that the trends with date are not consistent but there are some similarities with latitude and thus the anomalous AO/GBT observations are more likely related to their location than a transient phenomenon.

We added ticks indicating every degree of latitude.

I. 170-185:

The author clearly identify the "anomalous reflections" only at this point of the text ... in my opinion they should be identify before.

We added several paragraphs and a figure to the start of the results section to quickly and clearly introduce the anomalous reflections.

> In Fig. 2 please put anomalous reflections measurements in another color.

We added several paragraphs and a figure to the start of the results section to quickly and clearly introduce the anomalous reflections. Since the anomalous observations are now identified well before this figure, we do not think it is necessary to highlight them here. We considered changing the color of the anomalous observations in this figure but found that it distracts from the main point of the figure (latitudinal vs. temporal dependence). A new figure, earlier in the results section, has been added whose main point is identifying the anomalous observations.

I. 87-196:

"Titan has dunes analogous to sand dunes on Earth except that the grains are likely composed of hydrocarbons (tholins) rather than silicates (e.g., ref. 15). Dune fields cover millions of square kilometers on Titan, primarily in its tropical region, and are one of Titan's main terrain units. The AO/GBT specular reflections are from Titan's tropical region. From this cursory perspective, it is tempting to relate the specular reflections to the dunes or interdune areas (which are brighter than the dunes in Cassini altimetry observations 16). On closer inspection, however, there is no correlation. Seven AO/GBT locations are clearly dune/interdune regions in RADAR images and their AO/GBT maximum-NRCS range from 0.56-1.09. From Figure 2, these NRCS values are not anomalous and do not include the specular reflections of interest. Thus, the dune fields are decidedly ruled-out as the source of the AO/GBT specular reflections."

> Very well, but why dune fields should be considered as very efficient reflectors? Since they are pretty rough ... a priori they should be bad reflector! What is the underlying physical process/scenario assumed by the authors?

The dunes fields are rough at the large scales of individual dunes (~1 km). The dunes may also be rough at the scale of individual grains (~1 mm). However, the dunes and interdunes may be smooth at the scale of the AO/GBT observation wavelength ($\lambda=12.6$ cm). A dune's surface may be tilted but the tilted surface itself could be smooth.

Added the sentence: "Although the dune fields have rough surfaces at the scale of individual dunes (≈ 1 km), and possibly also at the scale of individual grains (≈ 1 mm), the surface of each dune and interdune may be smooth, as on the Earth, at the scale of the AO/GBT observation wavelength ($\lambda=12.6$ cm, ref. ¹⁶)." ."

Fig. 3: is the most convincing argument up to this point. The correlation between AO/GBT reflection and 5 microns bright zones identified by MacKenzie et al. is striking.

We agree that the correlation is striking!

I. 226-227: How the radar brightness of these possible paleolakes/paleoseas compares with the radar brightness of river dry beds proposed by Le Gall et al. (2010)?

The riverbeds discussed in Le Gall et al. (2010) are brighter than Hotei and Tui Regions in Cassini RADAR images (recall that all three regions are brighter than average in RADAR images). One can get a sense of the relative brightness from Fig. 2B (previously Fig. 1B). There are several riverbeds toward the right side of the figure; the riverbeds in the figure are not the specific riverbeds shown in Le Gall et al., (2010) but we verified that they have a similar brightness. Hotei Regio is the area around the largest red dot.

Note that the riverbeds are brighter in Cassini RADAR images, which are observations at non-nadir incidence angles, typically ~20-30 degrees incidence. But the riverbeds may not be brighter in nadir observations. Figure 1B shows that there are a couple of AO/GBT observations with subradar points near the riverbeds but those AO/GBT observations are not anomalously specular like the observations with subradar points on Hotei Regio.

I. 254: "The dielectric constant of a surface depends on its composition (and other parameters) ... ": what are these "other parameters"? Chemical composition is not the main parameter?

Changed to: "The dielectric constant of a surface depends on its composition (and other parameters, e.g., porosity) and Hotei/Tui have a different composition than their surroundings; recall that they are spectrally distinct, particularly in Titan's 5-micron atmospheric window (e.g., ref. ²¹)." ."

I. 266-267: "There are two noteworthy exceptions to the strong correlation of specular AO/GBT observations to Hotei/Tui. The first has a subradar point at $\approx 47^\circ\text{W}$, 13°S , ... ": please show this point with a label, an arrow ... in one map.

Added an arrow and label to map in Fig. 4 (previously Fig. 3).

I. 275-275: “The second exception has a subradar point at $\approx 163^\circ\text{W}$, 24°S and was acquired on 11/3/2000.” : idem, label in a figure/map.

Added an arrow and label to map in Fig. 4 (previously Fig. 3).

I. 299-304: “Using Titan as a solar system ground-truth in the search for oceans on exoearths suggests that, to identify liquids by specular reflections, a stringent definition of specular should be used. We recommend a definition based on the coherence of the reflected electromagnetic waves rather than definitions based on combinations of relative brightness, incidence angle, location, and/or polarization; reflections that are specular in the latter senses are quasi-specular.”

> For “exo-ocean-planet”, the integration time of an electromagnetic signal could be quite long compared to timescale associated to waves, this could significantly alter the coherence of the signal. But, the author are right encouraging scientific discussions about what could be the best criterion for the identification of specular reflection.

We agree that the observational parameters will differ significantly from that for Titan.

Reviewer #3

Thank you for the review, it has improved the flow and readability of the manuscript.

I do think the results presented in this manuscript would be of interest to the planetary community. However, to improve the impact factor, I recommend a rewrite of the manuscript to clean up some problems. The main issue is the way the manuscript is written – it is highly disconnected to a point that it is difficult to read. Grammatically, the manuscript is fine. However, due to the disjointed nature of the writing, the manuscript is very hard to follow and therefore not as easy to interpret. For example, consider the sentence starting with “Thick haze and abundant methane...” on lines 31 and 32. The sentence is out of place and has nothing to do with the sentences around it. My guess would be that the author is trying to tell a story about the aerosol, which obscures Titan’s lower atmosphere and surface at visible wavelengths – this is justification for going to longer wavelengths. But, this is well known and as written, it is highly disconnected and out of place, which is true for much of the manuscript. It almost reads like random thoughts were placed together, which borders on being incoherent. This type of disconnect in the writing appears throughout the entire manuscript.

We removed the sentence beginning with “Thick haze and abundant methane”.

We substantially rewrote and reorganized the manuscript to improve its flow.

The structure of the manuscript is also rather odd. It begins with a two paragraph Abstract, followed by a very long Result section with no subsections, and ends with a Discussion section. As written, I’m not convinced that ground-based AO/GBT observations are indeed specular. This impact factor is simply not in the as-written manuscript. A major rewrite is needed especially if published in a higher quality journal.

We added the section title “Introduction” to the introduction section, which separates the abstract and results section. We also expanded this section. We added subsection titles within the results section.

The word *specular* is used by various investigators with slightly different meanings in different contexts. Some of the AO/GBT observations are specular in the sense that they have a very narrow and very bright peak in the central bin of their Doppler spectrum. We added several paragraphs and a figure to the start of the results section to show these observations and demonstrate that they are anomalously specular. To avoid ambiguity, especially in regard to detection of liquid surfaces, we recommend a definition for specular that is based on the coherence of the reflected electromagnetic waves. This is discussed in detail in the manuscript.

As noted above, we did a major rewrite and reorganization to improve the manuscript’s structure and flow.

In the Results section, when you call out Fig. 1, make sure to state in the text (just like you do in the figure caption) that these are maps from the Cassini RADAR instrument. Stating just a “Cassini map” is careless. Also, when were the RADAR swaths acquired? I would add the time and Cassini flybys in the text and captions, and also do the same for the VIMS data shown.

We removed “Cassini” from the sentence in the text. We do not call it a “Cassini RADAR map” because at this point we are introducing the ground-based observations and have found that using “RADAR” here can create confusion, for readers that are not familiar with the Cassini mission, between the ground-based and Cassini observations.

The Cassini RADAR observations were acquired over the course of the Cassini mission from 2004-2017 and we have used the complete dataset. The observations were acquired on >50 different flybys, which is far too many to include in the captions. Instead, we added a table of the flyby dates to the supplementary material.

If you are going to dive into four very long explanations as to how Figure 1 is different than Figure 17 in Black et al., then you need to first introduce what Black et al. was reporting. Instead, I'd add a subsection and discuss how this work effort compares to other efforts. And, make sure to show more figures, especially talk to the ones you keep referring to in other publications (like those of Black et al.).

We added a subsection at the start of the results section to introduce previous work and explain how this work compares. The new subsection includes a new figure with multiple panels, including relevant figures from Black et al. We also shortened the explanation of how Fig. 1 differs from Fig. 17 in Black et al.

To improve the manuscript, I would really rethink the figures. Fig. 1B doesn't add much. Fig. 2 B and D don't add much either – A and C carry the impact factor. Do you really need Fig. 3A since you have Fig. 1A? Fig. 3B,C, and D are really good.

We carefully considered the figures and modified some of them but decided that they are all worth keeping. Fig. 1B (now Fig. 2B) highlights a correlation between areas with high NRCS in AO/GBT observations and areas with high NRCS in Cassini altimetry observations. This is an important point for the paleolake/paleosea hypothesis. Fig. 2B and 2D (now Fig. 3B and 3D) demonstrate the lack of temporal correlation between high NRCS AO/GBT and high NRCS altimetry observations. This is an important point for the transient liquids hypothesis. Fig. 3A (now Fig. 4A) demonstrates that the high NRCS AO/GBT observations are strongly correlated to the five-micron-bright regions Hotei Regio and Tui Regio. This is a major result of the paper and was identified as “striking” by another reviewer.

Reviewers' comments:

Reviewer #1 (Remarks to the Author):

The authors satisfactorily reworked the manuscript. It is well written and will certainly be of high interest to the scientific community. I gladly recommend this work for publication.

Reviewer #2 (Remarks to the Author):

The authors have made a nice effort in order to clarify their manuscript. I have only a few minor things:

Fig 1A-1E: the width of the echo should be indicated by a thicker line, in a well visible color (not black please).

L. 81-82: is Fig. 1E an example of an anomalous specular radar reflection (ASRR)? In addition, the two 'group' of ASRR in Fig. 1F could be graphically particularized using colors, labels, ...

I. 104-105: 'The maximum-NRCS depends on only the central Doppler bin and thus has a greater weighting from the terrain at the subradar location.': perhaps a ref. (if it is possible under NatCom format) ?

Fig. 4: please indicate the location of B. and D on map A.

I. 396: Data availability, please mentioned also the tool 'Titan Trek' <https://trek.nasa.gov/titan/>

Q.: by curiosity : how the NRCS of ASRR compares with NRCS in central part of evaporite deposits proposed by Barnes et al. (2009,2011)?

Reviewer #3 (Remarks to the Author):

The Root of Specular Reflections from Solid Surfaces on Saturn's Moon Titan

The manuscript definitely flows better than before (and has improved). I think it could be improved even more, especially the order of some sections (this will help with the flow).

Abstract:

L20-21: since hydrology means water, you shouldn't refer to Titan's methane cycle as "methane hydrologic cycle." Instead, you could say "Titan's tropospheric methane cycle" and then talk about how it is akin to Earth's tropospheric water cycle." Since the role of CH₄ at altitudes above Titan's tropopause are very different than Earth's water, you should restrict your water cycle and methane cycles comparisons to the two tropospheres.

L39: You mention that Cassini observations (RADAR and VIMS I presume) did not observe liquid reservoirs at the same regions that are anomalously specular to AO/GBT. I'm curious to know if your detections overlap spatially with where Cassini VIMS/ISS observed low latitude tropospheric methane rain/clouds. And also with ground-based AO Keck and Gemini CH₄ cloud observations (like those from Schaller et al.)?

Introduction:

L34-41: after southern tropical region, add the range in latitudes—equator to 30S? You could also say "low southern latitudes." Regardless, just be clear on the meridional (and zonal) locations. This is important b/c I'm immediately thinking about the season and the atmospheric dynamics, and if methane precipitation would be expected for the latitude and time of your ground-based observations.

After Introduction, I would expect a section on "Observations." Diving right into Results is rather odd since the reader doesn't know much about the observations, the spatial resolution, footprint size on Titan's disk, how this compares with Cassini RADAR and VIMS (if at all) starting July 2004 (Cassini orbit insertion), etc. In time, you overlap between 2004 and 2008 so this is the place to discuss this.

This would also be a good place to discuss an overlaps with the Keck/Gemini AO observations -- these are the ones that routinely monitored CH₄ tropospheric cloud activity from Earth.

Results

The first paragraph is about the quality of data, the fact that Titan's disk is spatially unresolved, what echo spectra look like, etc. This is info on the observations. It is not results. Same goes for most of the panels in Fig. 1.

L178: It took me a while to actually find the horizontal black lines. Consider increasing their values so the lines stand out better. Or, you could just say (in words) that between xx and xx Hz represents the width of the echo and outside of this range is just noise (if that is what you mean). Again, put all of this type of discussion in "Observations" since you are simply discussing the quality of the data.

L191-192: You didn't consider ground-based Keck and/or Gemini observations using adaptive optics that did show tropospheric clouds in the tropics during this time period. You need to detail these and address if there are any correlations (e.g., Schaller et al., 2009).

L210:213: What are you trying to say here? That temporal changes are an unnecessary hypothesis? This could be interpreted as somewhat aggressive. Instead, just state that temporal changes are unlikely to explain the AO/GBT ASRR.

L304: remove "..., they are the source of the ASRR." In the beginning of the sentence you state this so you don't need to say it again.

L371: Why is there a Methods section following Discussion? I would delete "Methods" and then move the content to the "Observations" section that should follow the Introduction section. The manuscript would then end with Acknowledgements/Author Contributions.

I really encourage the authors to create an Observation section following Introduction. Then, everything about your ground-based observations, combined with Cassini RADAR/VIMS, and Keck/Gemini observations would be revealed early on. I think this'll make the Results section more concise and more impactful.

Reviewer #1 (Remarks to the Author):

The authors satisfactorily reworked the manuscript. It is well written and will certainly be of high interest to the scientific community. I gladly recommend this work for publication.

Thank you for your service and remarks.

Reviewer #2 (Remarks to the Author):

The authors have made a nice effort in order to clarify their manuscript. I have only a few minor things:

Fig 1A-1E: the width of the echo should be indicated by a thicker line, in a well visible color (not black please).

The thickness of the line was increased and its color was changed from black to green.

L. 81-82: is Fig. 1E an example of an anomalous specular radar reflection (ASRR)? In addition, the two 'group' of ASRR in Fig. 1F could be graphically particularized using colors, labels, ...

Changed:

Observations with anomalously high peaks are the AO/GBT anomalously specular radar reflections (ASRR).

to:

Observations with anomalously high peaks (e.g., **D** and **E**) are the AO/GBT anomalously specular radar reflections (ASRR).

I. 104-105: 'The maximum-NRCS depends on only the central Doppler bin and thus has a greater weighting from the terrain at the subradar location.' : perhaps a ref. (if it is possible under NatCom format) ?

Changed to:

The maximum-NRCS depends on only the central Doppler bin and thus has a greater weighting from the terrain at the subradar location².

Fig. 4: please indicate the location of B. and D on map A.

The locations shown in B and D correspond to the two largest red dots in A. Adding annotations in A overcrowds those areas of the figure since they already have the Hotei and Tui boundaries, the Hotei and Tui labels, and several large AO/GBT dots. Instead we changed:

B The 18 km subradar track of the highest maximum-NRCS (and most specular in Black et al.²) AO/GBT observation, located in Hotei Regio.

to:

B The 18 km subradar track of the highest maximum-NRCS (and most specular in Black et al.²) AO/GBT observation, located in Hotei Regio (largest red dot in Hotei Regio in panel A).

and changed:

D The 18 km subradar track of the second highest maximum-NRCS (and second most specular in Black et al.²) AO/GBT observation, located in Tui Regio.

to:

D The 18 km subradar track of the second highest maximum-NRCS (and second most specular in Black et al.²) AO/GBT observation, located in Tui Regio (largest red dot in Tui Regio in panel A).

I. 396: Data availability, please mentioned also the tool 'Titan Trek' <https://trek.nasa.gov/titan/>

Added: Titan Trek is a NASA web-based portal with a browsing tool that allows for easy viewing and layering of Cassini RADAR images as well as other Cassini data of Titan; it is available at <https://trek.nasa.gov/titan/>.

Q.: by curiosity : how the NRCS of ASRR compares with NRCS in central part of evaporite deposits proposed by Barnes et al. (2009,2011)?

The sediment and evaporite deposits proposed in Barnes et al. (2009, 2011) are located in Titan's polar regions. Unfortunately, the Earth-Titan geometry precludes measuring the subradar NRCS of these regions with AO/GBT (only latitudes from ~7 - 27 S can be measured; also see lines 121-127).

Thank you for your service and remarks.

Reviewer #3 (Remarks to the Author):

The Root of Specular Reflections from Solid Surfaces on Saturn's Moon Titan

The manuscript definitely flows better than before (and has improved). I think it could be improved even more, especially the order of some sections (this will help with the flow).

Abstract:

L20-21: since hydrology means water, you shouldn't refer to Titan's methane cycle as "methane hydrologic cycle." Instead, you could say "Titan's tropospheric methane cycle" and then talk about how it is akin to Earth's tropospheric water cycle." Since the role of CH₄ at altitudes above Titan's tropopause are very different than Earth's water, you should restrict your water cycle and methane cycles comparisons to the two tropospheres.

Changed:

Titan has a methane hydrologic cycle with clouds, rain, rivers, lakes, and seas; it is the only world known to presently have a hydrologic cycle akin to Earth's water cycle.

to:

Titan has a methane cycle with clouds, rain, rivers, lakes, and seas; it is the only world known to presently have a volatile cycle akin to Earth's tropospheric water cycle.

L39: You mention that Cassini observations (RADAR and VIMS I presume) did not observe liquid reservoirs at the same regions that are anomalously specular to AO/GBT. I'm curious to know if your detections overlap spatially with where Cassini VIMS/ISS observed low latitude tropospheric methane rain/clouds. And also with ground-based AO Keck and Gemini CH₄ cloud observations (like those from Schaller et al.)?

The ASRR are not from the locations where Turtle et al. (2011; ISS) and Barnes et al. (2013; VIMS) reported clouds and rain. Some AO/GBT subradar locations were in this region, however, those observations are not anomalously specular. Importantly, these clouds and rain occurred in 2010, after all of the AO/GBT observations.

Some clouds in Schaller et al. (2009) are near the ASRR regions, however, those clouds also occurred after all of the AO/GBT observations. The clouds in Roe et al. (2005), Schaller et al. (2006), de Pater et al. (2006), and Hirtzig et al. (2006) were all at latitudes much further south than the AO/GBT observations.

This is discussed at the start of the Transient Liquids Hypothesis subsection; see also response to remark about lines 191-192.

Introduction:

L34-41: after southern tropical region, add the range in latitudes—equator to 30S? You could also say "low southern latitudes." Regardless, just be clear on the meridional (and zonal) locations. This is important b/c I'm immediately thinking about the season and the atmospheric

dynamics, and if methane precipitation would be expected for the latitude and time of your ground-based observations.

Changed:

Anomalously specular radar reflections (ASRR) from the southern tropical region of Saturn's moon Titan (Saturn and Titan have a solar obliquity of $\approx 27^\circ$) were observed with the Arecibo Observatory (AO) and Green Bank Telescope (GBT) from 2000-2008 and interpreted as evidence for liquid surfaces^{1,2}.

to:

Anomalously specular radar reflections (ASRR) from the southern tropical region of Saturn's moon Titan (equator to $\approx 27^\circ\text{S}$, Saturn and Titan have a solar obliquity of $\approx 27^\circ$) were observed with the Arecibo Observatory (AO) and Green Bank Telescope (GBT) from 2000-2008 and interpreted as evidence for liquid surfaces^{1,2}.

Also changed lines 124-125:

The AO-Titan geometry further limits the subradar locations to a latitude range of $\approx 20^\circ$ within Titan's southern tropical region.

to:

The AO-Titan geometry further limits the subradar locations to a latitude range of $\approx 20^\circ$ within Titan's southern tropical region (i.e., from $7\text{-}27^\circ\text{S}$).

After Introduction, I would expect a section on "Observations." Diving right into Results is rather odd since the reader doesn't know much about the observations, the spatial resolution, footprint size on Titan's disk, how this compares with Cassini RADAR and VIMS (if at all) starting July 2004 (Cassini orbit insertion), etc. In time, you overlap between 2004 and 2008 so this is the place to discuss this. This would also be a good place to discuss an overlaps with the Keck/Gemini AO observations -- these are the ones that routinely monitored CH₄ tropospheric cloud activity from Earth.

We considered this suggestion, however, the format for Nature Communications original research articles requires sections with the following order and titles: Abstract, Introduction, Results, Discussion, Methods. The journal's format policies do not permit an Observations section.

Subheadings within the Results (and Methods) section are also required but the titles of the subheadings are not fixed by the journal. Subheadings are not allowed in the other sections. We use the subheadings to help the reader follow the flow of the Results section. We think the subheadings of "Arecibo Observatory and Green Bank Telescope Anomalously Specular Radar

Reflections” and “Arecibo Observatory, Green Bank Telescope, and Cassini Observations” are more descriptive and helpful to the reader than a subheading of “Observations.” Although an observations section is not allowed, we provide the relevant details of the observations early in the Results section and take care to explain where we are describing previous observations as opposed to new results.

We considered moving some of the description of the observations that is in the Results section to the Introduction section, but figures in the Introduction section are discouraged by the journal. Also, Fig. 1 and 2 do contain new results. As such, we think that keeping these figures and their associated text at the beginning of the Results section, with descriptive subheadings, is the best available option.

Results

The first paragraph is about the quality of data, the fact that Titan’s disk is spatially unresolved, what echo spectra look like, etc. This is info on the observations. It is not results. Same goes for most of the panels in Fig. 1.

See response above regarding Observations section and organization of Nature Communications articles.

L 78: It took me a while to actually find the horizontal black lines. Consider increasing their values so the lines stand out better. Or, you could just say (in words) that between xx and xx Hz represents the width of the echo and outside of this range is just noise (if that is what you mean). Again, put all of this type of discussion in “Observations” since you are simply discussing the quality of the data.

The thickness of the line was increased and its color was changed from black to green.

L191-192: You didn’t consider ground-based Keck and/or Gemini observations using adaptive optics that did show tropospheric clouds in the tropics during this time period. You need to detail these and address if there are any correlations (e.g., Schaller et al., 2009).

Ground-based results, including Schaller et al. (2009), are included in Rodriguez et al. (2011) and Turtle et al. (2018) and were considered. We cited these summary Titan cloud papers rather than each individual Titan cloud paper.

Changed:

Many other clouds were observed on Titan^{13,14}, but they have not been associated with subsequent surface changes.

to:

Many other clouds were observed on Titan by both Cassini and Earth-based telescopes (e.g., ref. ¹³⁻¹⁶), but they have not been associated with subsequent surface changes.

Added references to Schaller et al. (2009) and Brown et al. (2010), which also summarizes cloud observations during Cassini's prime mission.

Note that ground-based detections of clouds in the tropics have not been associated to subsequent surface changes. We agree that the ground-based observations are an important dataset and pertinent to Titan's methane cycle. However, since they are not known to result in surface changes, discussing those observations in detail would be tangential to the ASRR, which are echoes from the surface. Furthermore, a hypothesis unrelated to transient liquids and the seasonal methane cycle is favored in this paper, so extending the discussion about clouds would belabor a hypothesis that is ultimately disfavored.

Also note that the clouds reported in Schaller et al. (2009) were detected after all of the AO/GBT observations.

L210:213: What are you trying to say here? That temporal changes are an unnecessary hypothesis? This could be interpreted as somewhat aggressive. Instead, just state that temporal changes are unlikely to explain the AO/GBT ASRR.

Changed:

Temporal changes are therefore an unlikely and unnecessary hypothesis.

to:

In summary, temporal changes are an unlikely hypothesis.

L304: remove "..., they are the source of the ASRR." In the beginning of the sentence you state this so you don't need to say it again.

Done.

L371: Why is there a Methods section following Discussion? I would delete "Methods" and then move the content to the "Observations" section that should follow the Introduction section. The manuscript would then end with Acknowledgements/Author Contributions.

A Methods section immediately after the Discussion section is required for all Nature Communications research articles according to the guide to authors. See also response to remark about an Observations section after the Introduction section.

I really encourage the authors to create an Observation section following Introduction. Then, everything about your ground-based observations, combined with Cassini RADAR/VIMS, and

Keck/Gemini observations would be revealed early on. I think this'll make the Results section more concise and more impactful.

We thank the reviewer for this thoughtful suggestion to improve the flow, however, an Observations section is not allowed in Nature Communications articles. See response above regarding an Observation section and rules for sections in Nature Communications articles.

Thank you for your service and remarks.

REVIEWERS' COMMENTS:

Reviewer #2 (Remarks to the Author):

I found the manuscript ready to be accepted.

This is a nice paper.

The Authors really made a great job!

Reviewer #3 (Remarks to the Author):

I appreciate the authors taking the time to address all of the reviewer suggestions. The manuscript has significantly improved and I recommend it for publication.